# Effects of Pre-Fire Vegetation on the Post-Fire Plant Community Response to Wildfire along a Successional Gradient in Western Juniper Woodlands

Eva K. Strand * and Stephen C. Bunting

Department of Forest, Rangeland, and Fire Sciences, University of Idaho, Perimeter Drive 875 MS 1135, Moscow, ID 83844-1135, USA
* Correspondence: evas@uidaho.edu; Tel.: +1-208-885-5779

**Abstract:** Western juniper was often historically restricted to fire refugia such as rocky outcrops but has since Euro-American settlement expanded into areas previously dominated by sagebrush steppe. Wildfires in developed woodlands have been rare. In 2007, the Tongue-Crutcher Wildland Fire burned 18,890 ha in southwestern Idaho along a woodland development gradient, providing unique research opportunities. To assess fire effects on vascular plants, field data were collected in 2012/2013 and 2019/2020. Species richness was uniform along the sere, while species diversity declined in late woodland stages attributed to juniper dominance. The greatest changes in species composition following fire occurred in later woodland development phases. Herbaceous vegetation increased following fire, but sagebrush cover was still lower in burned plots 12–13 years post-fire. Many stands dominated by juniper pre-fire became dominated by snowbrush ceanothus post-fire. Juniper seedlings were observed post-fire, indicating that juniper will reoccupy the area. Our research demonstrates resilience to fire and resistance to annual grasses particularly in early successional stages, which provides opportunities for fire use as a management tool on cool and moist ecological sites. Loss of old-growth juniper to wildfire underlines the importance of maintaining and provisioning for future development of some old growth on the landscape given century-long recovery times.

**Keywords:** mountain big sagebrush; burn severity; diversity; species turnover; secondary succession





## 1. Introduction

### 1.1. Extent of Sagebrush Steppe and Juniper Woodlands in the Great Basin

Pinyon–juniper woodlands are a dominant vegetation type throughout the semi-arid mountainous regions of the Great Basin, Columbia Basin and Colorado Plateau. Miller and Tausch [1] estimated that about 30 million ha are occupied by these woodlands in the western US. Before Euro-American settlement, it has been estimated that pinyon–juniper woodlands occupied less than 3 million ha [2]. Rapid expansion of the woodlands and infilling in established open woodlands [3] has occurred in the Euro-American period after 1800. The northwestern woodlands are dominated by western juniper (*Juniperus occidentalis* Hook var. *occidentalis*) and occupy 3.6 million ha, primarily in central and eastern Oregon, southwestern Idaho and northeastern California [1,4]. Causes of this expansion have been attributed to a number of factors including active and passive fire suppression, historic livestock grazing, fuel fragmentation (by roads, agricultural lands, etc.), climatic variation and change, and elevated $CO_2$ levels [4,5]. Following severe drought in recent years, increased mortality in pinyon pine (*Pinus monophylla* (Torr. and Frém.) and Utah juniper (*Juniperus osteosperma* (Torr.) Little) woodlands has been documented on warm dry sites in central Nevada [6]; however, diebacks have not been reported for western juniper.

*1.2. Woodland Development Phases, Mature Juniper and Juniper Expansion*

The successional transition of sagebrush shrub steppe into mature western juniper woodlands has been described in three phases [4]. In Phase I, the juniper trees are small, widely spaced and generally have only minor influences on ecological processes. Herbaceous and shrub components remain relatively intact and are minimally affected by the few and young juniper trees. In Phase II, the trees and understory shrub and herbaceous layers are co-dominant. The juniper trees are beginning to influence the site's composition and ecological processes. In Phase III, juniper dominates the site's composition and ecological processes. Many shrub steppe species have declined on the site and will continue to decline as the woodland develops. Mature woodland is dominated by juniper and sometimes pine species. Understory shrub and herbaceous species coverage is diminished from the shrub-steppe-dominated state due to competition with trees for light and other resources. Bare ground in the tree interspaces is common. The tree component often has a mixed age and size structure, but a majority of the site's plant coverage consists of mature trees. Total tree cover varies by region and soil type but usually exceeds 30%. The age of these mature trees often exceeds 400 years, and some individuals may be over 1000 years old. Landscapes are frequently a mosaic of different phases, mature western juniper and sagebrush steppe (Figure 1).

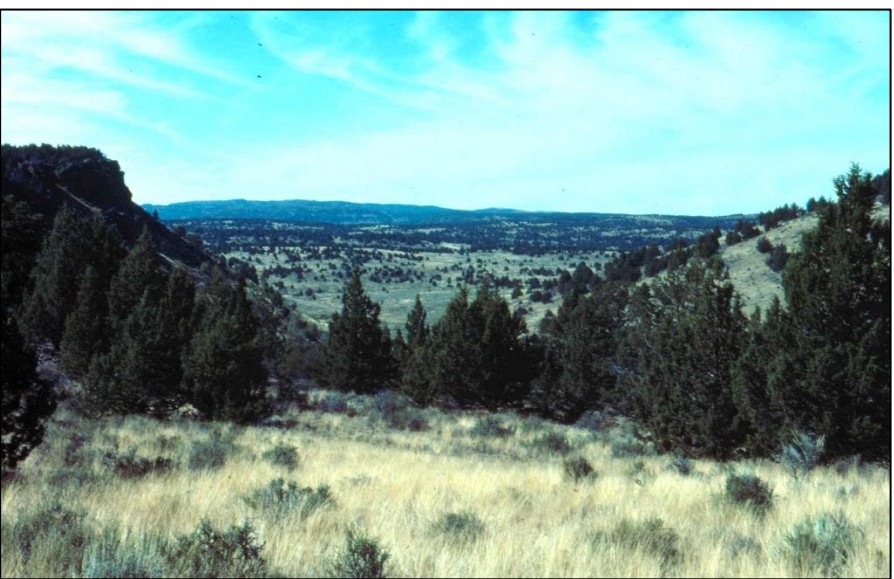

**Figure 1.** Overview of sagebrush steppe and western juniper woodland landscape in the Owyhee Mountains of southwestern Idaho.

In the last couple of decades, juniper removal treatments have been implemented with the goal of restoring sagebrush steppe vegetation and habitats [4] and reversing the ecological effects of juniper expansion. Prescribed fire or a variety of mechanical treatments, or a combination of the two, are common in early successional stages of juniper encroachment, but sagebrush steppe restoration treatments have also been implemented in Phase III woodlands [7]. Pre-settlement old woodlands are unique habitats and not a target for juniper removal. Old-growth juniper is estimated to make up less than 10% of the western juniper woodlands [8].

*1.3. Fire Regimes and Changes in Fuel Components*

Fire regimes describe spatial and temporal characteristics of wildfire in an ecosystem, for example fire frequency, burn severity, fire size and patchiness. Historical fire regimes provide context for the role of fire in an ecosystem and can help explain changes that are occurring within plant communities and landscapes as a result of deviations from historical fire regimes. Fire frequency in mountain big sagebrush was historically higher than in



other sagebrush communities because of higher productivity and more continuous fine surface fuels [9]. Fire frequency likely varied along moisture and temperature gradients and depending on the fire ignition and spread probability in adjacent cover types [10]. Reported historical fire intervals in mountain big sagebrush (*Artemisia tridentata* ssp. *vaseyana* [Rydb.] Beetle) are variable depending on region and adjacency. Miller and Tausch [1] estimated fire frequency in mesic mountain big sagebrush communities at 11 to 25 years based on analysis of adjacent fire scarred trees. Fire return intervals in mountain big sagebrush were suggested to be greater than 80 years in the mountain big sagebrush–bluebunch wheatgrass (*Pseudoroegneria spicata* [Pursh] A. Löve)–Thurber needlegrass (*Acnatherum thurberianum* [Piper] Barkworth) plant association with lower fuel loads and associated with large western juniper snags [11]. Burkhardt and Tisdale [12] suggested that mountain big sagebrush currently encroached by western juniper must historically have had a fire return interval < 50 years because western juniper trees less than 50 years of age are readily killed by fire, which would inhibit woodland expansion. Longer fire return intervals are suggested in the western juniper–mountain big sagebrush–western needlegrass plant associations with limited fine fuel production; presence of old western juniper trees indicated long fire-free periods [10]. Fires in mountain big sagebrush were likely small (<500 ha); large fires were infrequent [13]. In a systematic review, Baker and Shinneman [14] evaluated fire regime characteristics and the role of fire in pinyon–juniper woodlands. They found that most fires in pinyon–juniper woodlands burned at a high-severity infrequent fire regime and that low-severity surface fire was rare in developed woodlands because of low herbaceous and shrub fuel loads. Rather than a spreading surface fire, high-severity fires have been described as trees torching and ejecting large flaming embers composed of ropelike juniper bark strips, igniting neighboring trees and vegetation [15]. We assume that the pinyon–juniper woodlands described by Baker and Shinneman [14] are late successional of the Phase III or Mature woodland classification proposed by Miller et al. [4].

The fuel complex dramatically changes along the successional gradient from sagebrush steppe to mature juniper woodland. Fuel loading is primarily composed of herbaceous and small-diameter woody fuels (<25.4 mm) in sagebrush steppe and Phase I woodlands. As juniper becomes increasingly dominant on the site, the herbaceous fuel component declines [4,16]. The shrub component of the vegetation decreases, and given time, the woody shrub biomass transitions to surface fuels [16,17]. As the woodland matures, the fuel complex becomes increasing dominated by those fuels produced by the juniper trees, i.e., large-diameter woody fuels [16] and juniper litter and duff [18]. Large fuels and compacted litter and duff are less flammable but tend to smolder for longer periods of time, resulting in injury to plant tissue and reproductive structures [19]. These changes, in turn, affect many characteristics of the fires when they occur such as fire size and burn uniformity [20], flame length and rates of spread [16]. Thus, fires along the maturing woodland development gradient become less frequent and more severe when they occur [21,22].

Pre-fire plant composition and structure are important characteristics when evaluating post-fire plant succession and resilience, and resistance to invasive plants [10]. Resilience is defined as the capacity of ecosystems to reorganize and regain their structure, processes and functioning following a disturbance [23]. Resistance refers to an ecosystem's ability to resist community structural change following disturbance [24,25].

Few studies have reported on plant community response to wildfire along the mountain big sagebrush–western juniper woodland gradient. Increasing burn severity of wildfire has been documented along the woodland development gradient supported by both remote sensing and field reconnaissance [22]. A review of fire effects in mountain big sagebrush documents general impacts. Deep-rooted perennial grasses decline after fire but generally recover to pre-burn levels after 2–3 years, but recovery time depends on burn severity and post-fire environmental conditions [26,27]. Sagebrush is sensitive to fire and can take several years to decades to return to pre-fire cover levels [28]. Shrubs with the ability to sprout after fire or shrubs with soil-stored seeds can more rapidly return post-fire levels [27]. Western juniper trees under 1–2 m have thin bark and are readily killed by fire,



while older trees may survive low-intensity fire but are sensitive to crown scorch and killed in crown fires [29]. Wildfire can have severe effects on soil properties. Soil erosion increased by a factor of 20 following wildfire in western juniper woodlands and remained high in areas previously occupied by trees but was reduced in the inter-canopy because of post-fire herbaceous recruitment [30]. The 2007 Tongue-Crutcher Wildland Fire Complex that burned 18,890 ha across the gradient of western juniper woodland development phases and through mature juniper provided a unique opportunity to evaluate ecological relationships and plant community response to fire in this ecosystem.

Several metrics are used by ecologists for evaluating ecological differences and similarities between plant communities [31,32]. Species richness refers to the number of species in a community, while diversity also accounts for the proportional abundance of those species, for example, Shannon's Diversity Index [33]. Whittaker [34] partitioned diversity into alpha, beta and gamma components based on a spatial scale. Alpha diversity refers to the local level diversity at stands within a community type. Beta diversity represents the dissimilarity in species composition between sampling units and reflects the turnover or replacement of species between community types [32,34,35]. Gamma diversity accounts for the total number of species in the region under study. Biotic cover of live vegetation is another measure commonly used to evaluate revegetation in post-fire environments [36,37].

The overarching research question we address in this paper is how does pre-fire vegetation affect post-fire recovery in mountain big sagebrush steppe and western juniper woodlands following wildfire? Specifically, we conduct analyses to address the following questions: (1) What are the effects of wildfire on species diversity and species turnover along the woodland successional gradient? (2) How does the plant community composition change along the successional gradient and what are the effects of wildfire on the plant community? (3) Are there differences in cover of species functional groups along the successional gradient post-fire? (4) What are the dominant species in this plant community and how is the cover of those species affected by wildfire? Finally, we synthesize our results in a prediction of a long-term trajectory for stands that burn at different woodland development phases and for mature woodlands.

## 2. Methods

### 2.1. Site Description

Western juniper woodlands are native to eastern Oregon, northern California, northwestern Nevada and southwestern Idaho (Figure 2a), occupying approximately 3.6 million ha in the region. In 2007, the Tongue-Crutcher Wildland Fire Complex (TCWFC) burned through 18,890 hectares of sagebrush steppe and western juniper woodlands on Juniper Mountain, located in the southwestern part of the Owyhee Plateau in Idaho (42°20′39″ N; 116°50′32″ W).

Elevation within the study area ranged from 1300 m in the canyons to 2000 m towards the top of Juniper Mountain. Average annual precipitation ranges from 270 mm at the lowest elevation to 540 mm at the highest elevation within the fire perimeter [38]. The lowest average monthly temperature of −3.2 °C occurs in December, and the highest average monthly temperature of 21.5 °C occurs in July [38]. Parent material on Juniper Mountain resulted from a basaltic eruption and is composed of ashflow tuff and ignimbrite [39], with the dominant soil types being haplargids at lower elevations, haploxeralfs at mid-elevations and argixerolls towards the very top of the mountain [40]. Sampling was conducted in the northern part of the burn and adjacent unburned areas where a variety of woodland development phases were present. The ecological site sampled is in the Fastjet soil series classified as shrubby loam 330–406 mm precipitation, with dominant vegetation being big sagebrush (*Artemisia tridentata* Nutt.), Idaho fescue (*Festuca Idahoensis* Elmer) and antelope bitterbrush (*Purshia tridentata* [Pursh] DC.) [40]. The sampled area falls in the category of ecological sites classified as high resilience to fire and resistance to invasion by annual grasses [41].

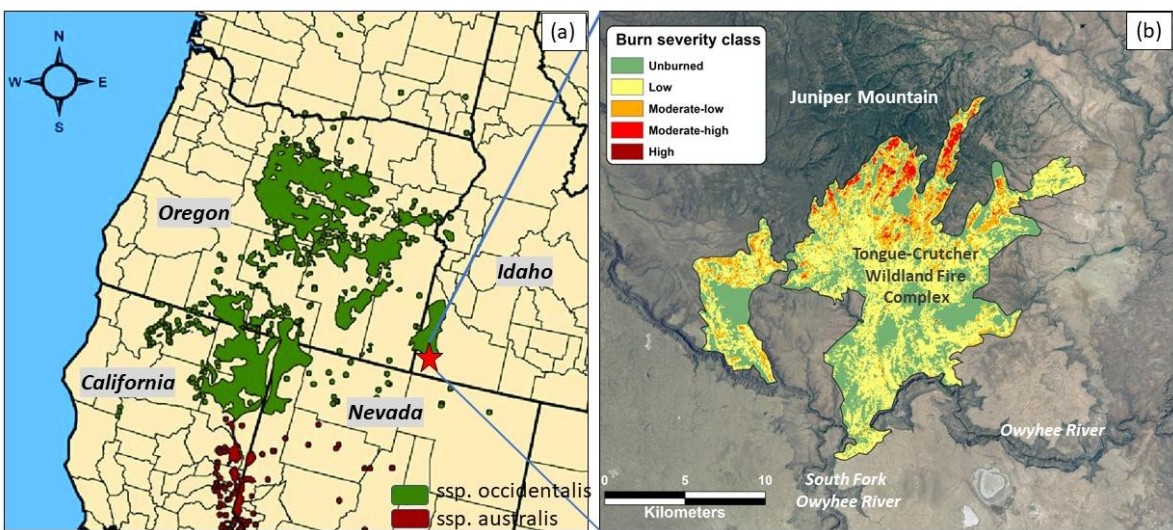

**Figure 2.** (**a**) Distribution of western juniper in the western United States. (**b**) Location of the 2007 Tongue-Crutcher Wildland Fire Complex on Juniper Mountain including burn severity classes within the fire perimeter.

Pre-fire vegetation was characterized by a mosaic of sagebrush steppe and western juniper (*Juniperus occidentalis* Hook. var. *occidentalis*) in various woodland development stages ranging from stand initiation woodlands (Phase I) to mature woodlands with trees upward of 500 years. The dominant sagebrush species within the sampled area was mountain big sagebrush (*Artemisia tridentata* ssp. *vaseyana* [Rydb.] Beetle) on deeper soils and little sagebrush (*Artemisia arbuscula* Nutt.) on shallow clay soils. Other xeric shrubs included rabbitbrush (*Chrysothamnus* Nutt. spp.) and antelope bitterbrush (*Purshia tridentata* [Pursh] DC). Curl-leaf mountain-mahogany (*Cercocarpus ledifolius* Nutt.) was widespread within the area, ranging in size from small shrubs to tall tree-like shrubs (>10 m tall). Mountain shrub species included mountain snowberry (*Symphoricarpos oreophilus* A. Gray), bittercherry (*Prunus emarginata* [Douglas ex Hook.] D. Dietr.) and chokecherry (*Prunus virginiana* L.). Snowbrush ceanothus (*Ceanothus velutinus* Douglas ex Hook.) was not common prior to the burn but present in a few openings. Native perennial grass species included bluebunch wheatgrass, Idaho fescue, needlegrasses (*Achnatherum* P. Beauv. spp.), oniongrass (*Melica bulbosa* Geyer ex Porter and J.M. Coult.) and Sandberg bluegrass (*Poa secunda* J. Presl). Native perennial forbs included arrowleaf balsamroot (*Balsamorhiza sagittata* [Pursh] Nutt.), lupine (*Lupinus* L. spp.), tapertip hawksbeard (*Crepis accuminata* Nutt.), desert parsley (Lomatium Raf. spp.) and buckwheat (*Eriogonum* Michx. spp.). Annual forbs were frequent but present at low cover levels because of their delicate stature, for example autumn willow-herb (*Epilobium brachycarpum* C. Presl), cryptantha (*Cryptantha* Lehm. ex G. Don spp.), blue-eyed Mary (*Collinsia parviflora* Lindl.), starwort (*Stellaria* L spp.), Douglas' knotweed (*Polygonum douglasii* Greene) and tiny trumpet (*Collomia linearis* Nutt.). Non-native forbs were rare except yellow salsify (*Tragopogon dubius* Scop.), which was common. Exotic annual grasses including cheatgrass (*Bromus tectorum* L.) and soft brome (*Bromus hordeaceus* L.) were present at low cover throughout the area.

The TCWFC started by lightning in two locations along the Owyhee River on July 6, following an unusually dry spring. Nearby weather stations in Rome, Oregon, and Murphy, Idaho, reported wind gusts of up to 22 ms$^{-1}$, temperatures over 38 °C and daytime relative humidity below 10%. The high-intensity fire burned through sagebrush steppe vegetation and juniper woodlands in a variety of woodland development stages (Phase I–III and Mature), initially spreading northward on Juniper Mountain, then spreading towards the east and west, exhibiting extreme fire behavior including spotting, torching and crowning. The fire partially burned an area named the Big Tree Mesa, an area characterized by open-canopy old pre-settlement woodlands surrounded by steep canyons both to the east and

west. These trees were likely some of the oldest western juniper in the region, implying the rarity of the conditions under which this wildfire burned. Suppression action began around 11 July with dozer lines implemented on the northern edge; the fire was mostly contained by 21 July. The burn severity index (delta Normalized Burn Ratio [21]) varied within the fire perimeter, with the highest burn severity index associated with late successional woodlands in the northern portion of the burned area (Figure 2b; [22]).

The remote land area is federally managed by the Bureau of Land Management (BLM) with a few parcels of private and Idaho State land. The study area is primarily located in two BLM grazing allotments. The Trout Springs allotment was largely unburned, while the Castlehead Lambert allotment's Between the Canyons pasture was mostly burned but contained unburned sections within the fire perimeter. Prior to the fire, target utilization was less than 50%. Within burned areas, the goal was 80% ground cover before continuing grazing. None of the burned pastures were grazed by cattle in 2008 or 2009, but grazing resumed in 2010 in the Trout Springs allotment. In the Between the Canyons pasture, where the bulk of the burned plots occurred, no livestock grazing occurred in 2010. Aside from cattle grazing in the late summer or fall, land use activities include hunting elk and deer, and recreation such as camping, hiking and all-terrain vehicle (ATV) use.

### 2.2. Field Sampling

Post-fire vegetation sampling to assess vascular plant community composition was conducted in 2012/2013 and again in 2019/2020. In 2012 and 2013 (5–6 years post-fire), 56 vegetation sampling plots were established, stratified by four woodland phases (Phase I–III and Mature) in areas that burned in the TCWFC (36 plots referred to as "burned") and in adjacent areas that did not burn in the fire (20 plots referred to as "unburned"). The same plots were resampled in 2019 and 2020 (12 and 13 years post-fire). In the following, we will refer to these sampling efforts as sampling period 1 (2012/2013) and sampling period 2 (2019/2020). At each plot location, two 25 m transects were established at random locations along a 20 m baseline. Percent canopy cover by species of all herbaceous vascular plants was recorded at each meter-mark within a 0.5 × 0.5 m quadrat, resulting in 50 quadrats total per plot. Percent rock was recorded within the quadrat considering the minimum size of a rock to be 5 × 5 cm. Plant species with less than 1% cover were referred to as "trace", and in later calculations, we assigned the value 0.1% to these species with low cover. Many of the species with low cover were, however, very frequent in our plots (present in many quadrats), and thus the sum cover for a plot may be exaggerated. The number of post-fire juniper seedlings was documented within the quadrats. Shrub cover by species was estimated along the same transects using the line intercept method. Ocular cover estimates were made for the remaining post-fire juniper (if any) for the plot as a whole. The location of each plot was recorded using a Garmin 76CSx Global Positioning System unit. Slope, aspect and elevation at the center of the baseline were recorded.

### 2.3. Statistical Analysis Methods

2.3.1. Species Richness and Diversity

Plant species data were summarized to the plot level at each location by averaging the plant cover in each of the 50 quadrats along the transects. Species richness was computed by adding all species of vascular plants present within the 50 quadrats within the plot plus the shrub species documented along the line intercept and juniper if present. We used Shannon's Diversity Index (H') as a measure of alpha diversity, the diversity at the plot level. Shannon's Diversity Index was calculated at the plot level using the following equation:

$$H' = -\sum_i^R p_i \ln(p_i) \tag{1}$$

where $p_i$ is the proportion of the $i$th species, and R is the species richness, the maximum number of species within a plot.

We tested for differences in species richness and H' between sampling period (sampling period 1 vs. 2), burn status (burned vs. unburned) and pre-fire woodland development phase (Phase I, II, III and Mature) and their interactions with a repeated measures Analysis of Variance (ANOVA). Prior to analysis, the variables were evaluated for normality using Q–Q plots and histograms. Pairwise comparisons were performed using Tukey's Honestly-Significant-Difference Test. All comparisons were conducted between plots of the same pre-burn woodland development phase.

To better understand the contribution of each woodland development stage to species diversity within the landscape, we also compared species richness and diversity between woodland development phases and the total landscape. For this analysis, we randomly selected five plots from each woodland development phase (unburned and burned) to avoid an uneven sample size since species richness is dependent on area sampled.

### 2.3.2. Plant Community Turnover and Composition

To assess differences in species turnover (a measure of beta diversity) between woodland phases along the successional gradient and to quantify the species turnover that occurred as a result of the fire, we computed the Sorensen (Bray–Curtis) distance measure of dissimilarity based on group averages [31]. Statistical differences in plant community composition were evaluated between woodland development phases for burned and unburned plots for the two sampling periods using the Multi-Response Permutation Procedure (MRPP) with pairwise comparisons [31]. The MRPP analysis generates two statistics, the *p*- and the A-value. The *p*-value represents the probability of a type I error under the null hypothesis that there is no difference between samples. The A-value is a measure of agreement between groups where A = 1 for complete within-group homogeneity, A = 0 when the heterogeneity within groups is equal to the expectation and A < 0 if there is less agreement within groups than expected by chance. Ecological communities are commonly A < 0.1, and within-group agreements with A > 0.3 are uncommon [31]. In the plant community analysis, we included those species with more than three non-zero values to reduce noise in the dataset introduced by infrequent species [31].

### 2.3.3. Functional Groups and Dominant Species

The vascular plant species documented in the plots were grouped in plant functional groups, and canopy cover by functional groups was calculated by adding the cover of the plant species within each group: annual grass, annual forbs, perennial grass, perennial forbs, shrubs and trees. Differences in functional group canopy cover between burned and unburned plots by woodland development phase and sampling period were tested with Student's *t*-test. We excluded annual forbs from this analysis because of the difficulty in estimating cover of species with <1% cover. Instead, we calculated and report frequency of frequent annual forbs. Frequency was calculated as the percentage of the quadrats that contained the plant.

The most common plant species in the dataset (>3% canopy cover across plots) were further evaluated. We tested for differences by woodland development phase by burn status (burned vs. unburned) between years and for the difference between burned and unburned plots by phase within the same year using Analysis of Variance. Pairwise differences between woodland development phases were evaluated with a post-hoc Tukey's Honestly-Significant-Difference test.

Juniper seedlings established post-fire were counted within the 50 quadrats along the transects and converted to seedling density (seedlings/$m^2$). Difference in mean seedling density between burned and unburned plots was tested with Student's t-test for each woodland development phase and for mature woodlands. We used SYSTAT [42] for statistical analysis and PCORD [43] for plant community analysis. Differences were considered significant at $p < 0.05$.

## 3. Results

### 3.1. Richness and Diversity

Differences in species richness and Shannon's Diversity Index (H') for sampling periods (TIME), the pre-fire woodland development phase (PHASE) and burn status (unburned vs. burned; BURN) were evaluated in a repeated-measures three-way ANOVA. Differences in species richness were observed by sampling period and pre-fire woodland development phase, while there was no difference in richness between unburned and burned stands (Table 1). None of the interaction terms were significant. For H', all variables (TIME, PHASE and BURN) were significant, while none of the interactions were (Table 1).

**Table 1.** Evaluation of changes in species richness per plot and Shannon's Diversity Index by sampling period (TIME; repeated measure), pre-fire woodland development phase (PHASE) and fire (BURN) and their interactions were conducted with a three-way repeated-measures ANOVA (df—degrees of freedom; MS—mean squares; F-ratio and *p*-value). Significant relationships ($p < 0.05$) are in bold font.

| Variable | Richness | | | | Shannon's Diversity Index | | | |
|---|---|---|---|---|---|---|---|---|
| | df | MS | F-Ratio | *p* | df | MS | F-Ratio | *p* |
| TIME | 1 | 611.02 | 24.34 | **<0.001** | 1 | 4.73 | 31.06 | **<0.001** |
| TIME*BURN | 1 | 26.27 | 1.05 | 0.311 | 1 | 0.23 | 1.50 | 0.226 |
| TIME*PHASE | 3 | 20.07 | 0.80 | 0.500 | 3 | 0.10 | 0.64 | 0.593 |
| TIME*BURN*PHASE | 3 | 48.18 | 1.92 | 0.139 | 3 | 0.06 | 0.38 | 0.766 |
| BURN | 1 | 30.46 | 0.80 | 0.375 | 1 | 3.66 | 14.22 | **<0.001** |
| PHASE | 3 | 217.36 | 5.72 | **0.002** | 3 | 8.82 | 34.21 | **<0.001** |
| BURN*PHASE | 3 | 6.61 | 0.17 | 0.914 | 3 | 0.15 | 0.57 | 0.635 |

Pairwise comparison analysis confirmed differences in species richness between early and late woodland development phases for sampling period 1, with decreasing richness along the woodland development gradient. The trend was similar for unburned and burned stands (Figure 3a). In the second sampling period, there was no difference in richness between the woodland development phases for unburned or burned stands (Figure 3b). No difference was observed in H' between unburned and burned woodland development phases in either sampling period, but H' was lower for the late-development stands (Phase III and Mature) in both sampling periods (Figure 3c,d).

Differences observed between sampling periods included increases in species richness for burned Phase I ($p = 0.008$), burned Phase III ($p = 0.001$) and burned Mature stands ($p < 0.001$). Increases in H' were observed for unburned Phase I ($p = 0.014$) and Phase III ($p = 0.024$) and burned Phase III ($p = 0.016$).

Total species richness within woodland development phases, the sum of all species within the woodland development phase rather than the average as displayed in Figure 3, was relatively constant along the sere for both sampling periods (Figure 4a,b). Little difference in richness was observed because of the burn. However, the total number of species sampled across the five randomly selected unburned and burned plots by phase was 124 during the first and 131 in the second sampling period. Diversity (H'), when assessed for all species within the woodland development phases, similar to the plot averages presented in Figure 3, was highest in the early woodland development phase and decreased along the sere. H' was higher in burned plots when compared to unburned plots of the same phase. H' was higher in the early woodland development phases. At the landscape scale, H' was higher in the burned plots due to the reduction in the dominance of juniper, which increased equitability between species.

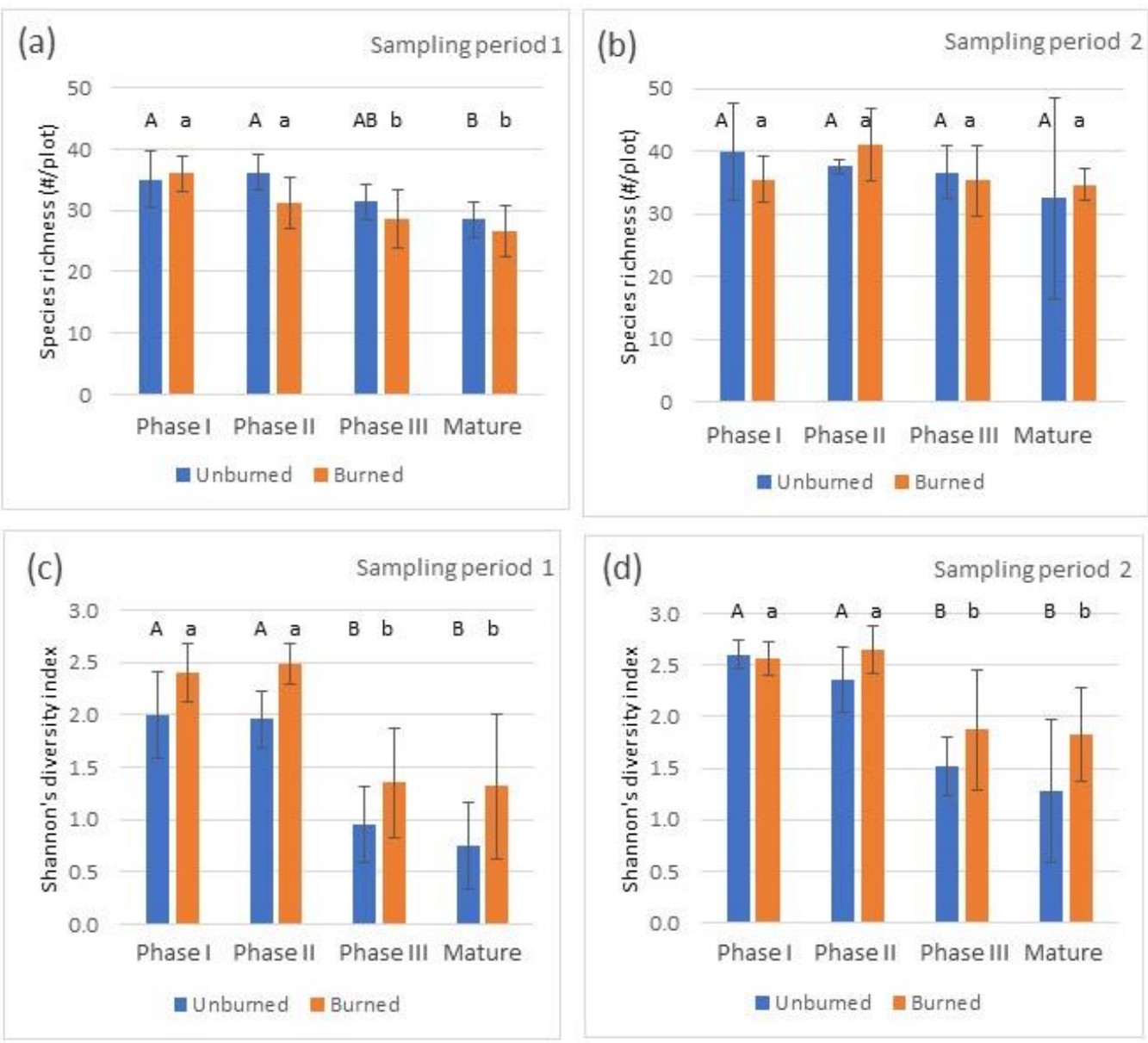

**Figure 3.** Species richness by woodland development phase in unburned and burned stands 5–6 years post-fire (**a**) and 12–13 years post-fire (**b**). Shannon's Diversity Index (H′) by woodland development phase in unburned and burned plots 5–6 years post-fire (**c**) and 12–13 years post-fire (**d**). Error bars represent standard deviation between plots within the same woodland development phase. Capital letters A and B indicates statistical difference ($p < 0.05$) between phases for unburned plots and lower-case a and b indicates difference between phases in burned plots. Bars with the same letter combination indicates there is no difference.

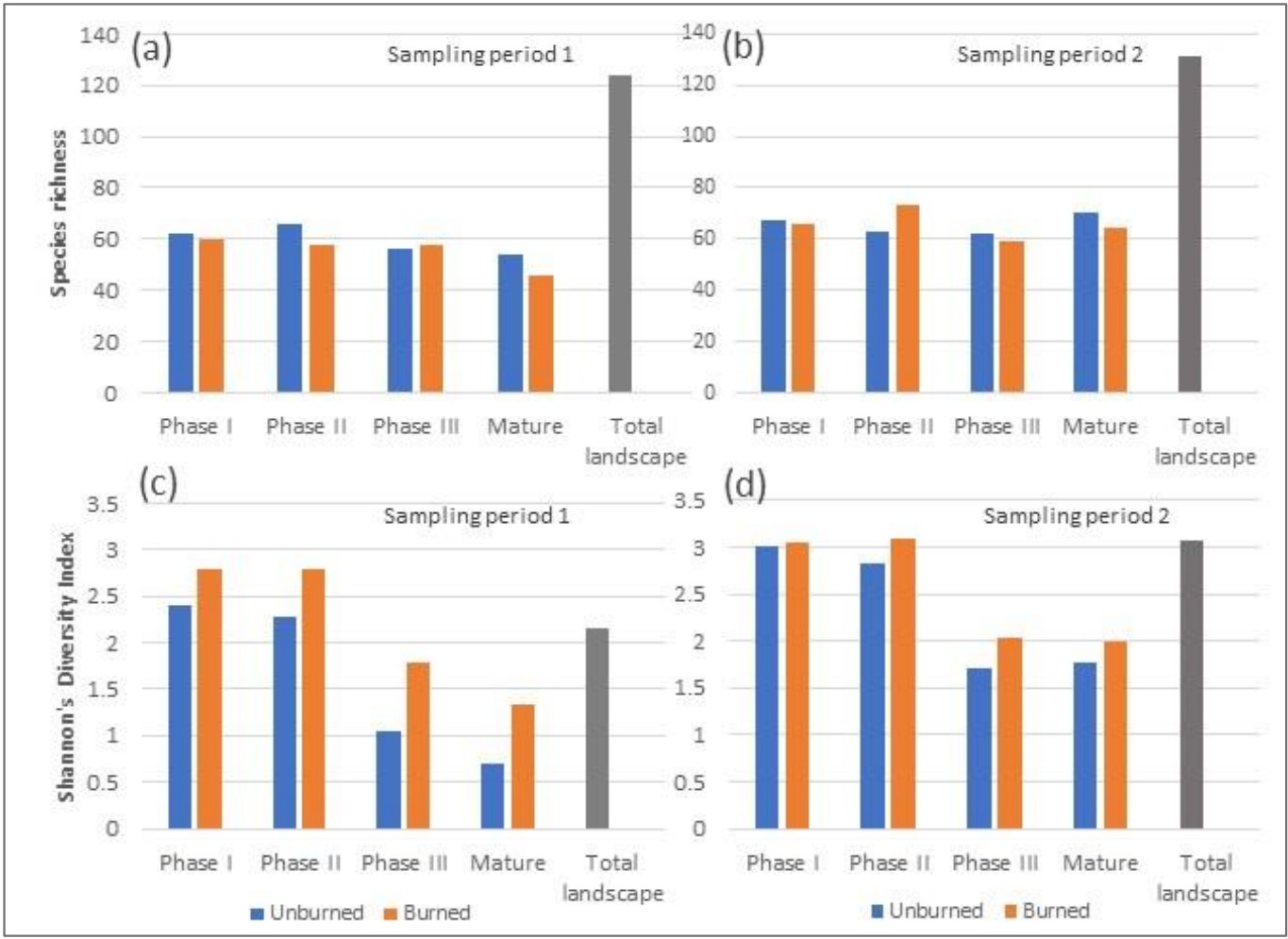

**Figure 4.** Total species richness (**a**,**b**) and diversity (**c**,**d**) for each woodland development stage (unburned and burned) and for the total landscape for the two sampling periods.

### 3.2. Plant Community Turnover and Composition

Differences in plant community composition were assessed using the Sorensen (Bray–Curtis) Dissimilarity Index (SDI) based on group averages [31]. The SDI ranges from zero if there is no difference in plant community composition (richness and abundance) to one if complete species turnover has occurred. Species turnover measured by the SDI was 0.414 between unburned Phase I and II, increased to 0.586 between Phase II and III, and was lower (0.075) between Phase II and Mature woodlands (Figure 5, top row). Turnover in species composition occurring as a result of fire increased along the successional gradient. The difference in the SDI between unburned and burned plots of Phase I was 0.419, while the SDI for Phase II, Phase III and Mature was 0.643, 0.745 and 0.945, respectively, when comparing unburned and burned plots 5–6 years after the fire (Figure 5). The species turnover between sampling periods (Time 1 and 2) was similar along the successional gradient, ranging from 0.407 to 0.490 (Figure 5).

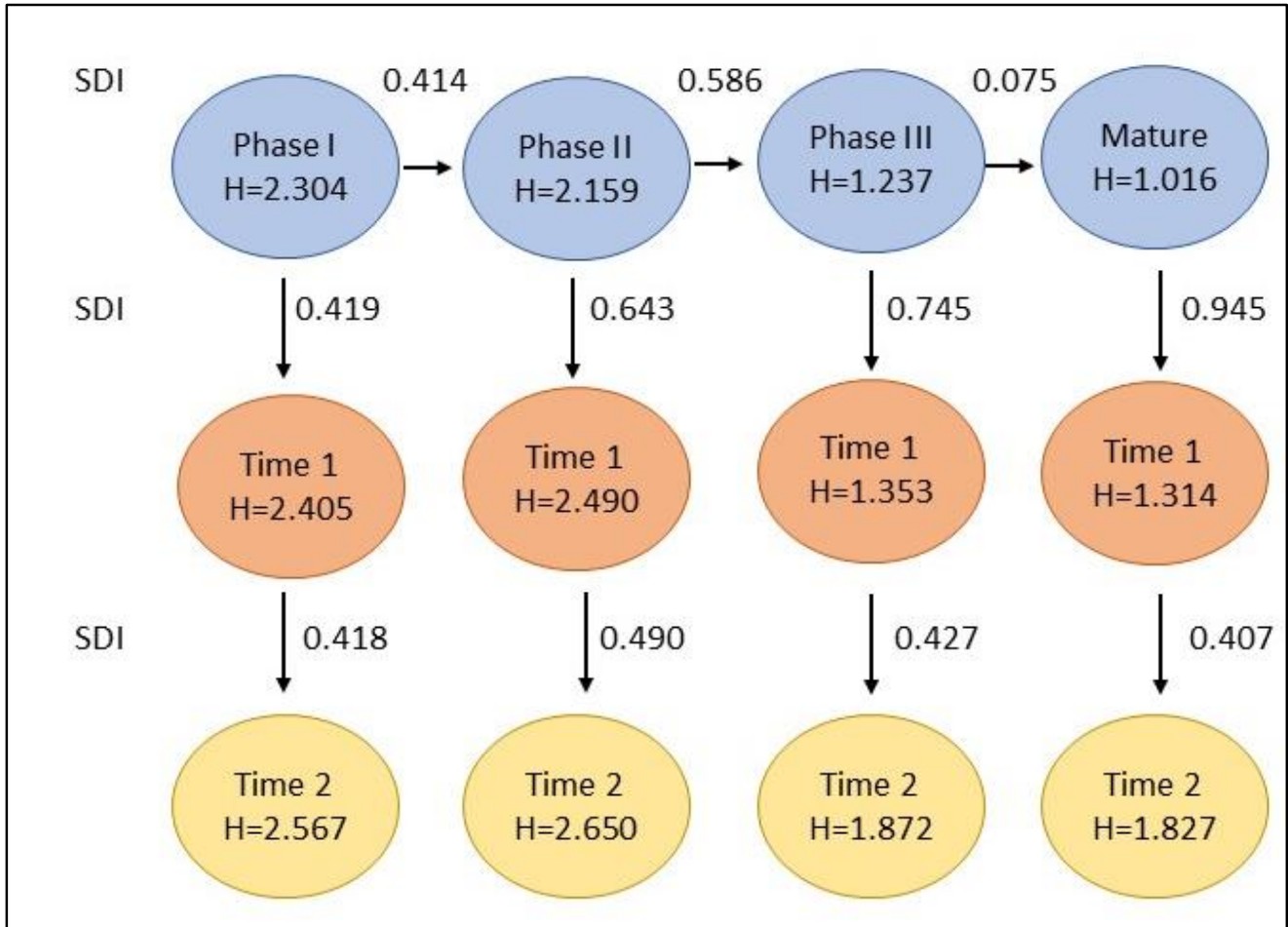

**Figure 5.** Average of Shannon's Diversity Index (H') for plots by woodland successional phase pre-fire (blue circles), 5–6 years after the wildfire (Time 1, orange circles middle row) and 12–13 years after the wildfire (Time 2, yellow circles bottom row). Sorensen's Dissimilarity Index (SDI) is a measure of dissimilarity or species turnover along the successional gradient and following the wildfire. High values represent high turnover of species. The largest observed change along the woodland development gradient is seen between Phase II and III. Species turnover because of the fire increased along the successional gradient with almost a complete replacement of the plant community in the Mature woodlands (SDI = 0.945).

Differences in plant community composition were evaluated with the Multi-Response Permutation Procedure [31]. Overall differences between woodland development phases were observed for both unburned and burned plots during both sampling periods (Table 2). No significant difference was observed between woodland development Phase I and II for the burned plots in either sampling period or for the unburned plots in sampling period 1. However, a statistical difference was observed between unburned Phase I and II plots in sampling period 2, but given the relatively low A value (A = 0.144), the ecological difference is small. Differences were observed between Phase I and III, Phase II and III, and Phase II and Mature for both unburned and burned plots in both time periods. No difference was observed between Phase III and Mature within unburned or burned plots for either sampling period.

**Table 2.** Differences in plant community composition between woodland development phases (PI, PII, PIII and Mature) for unburned and burned plots for the two time periods sampled. MRPP outputs are reported: T = test statistic; A = chance-corrected within-group agreement; *p* = probability. Probability values < 0.05 are in bold font. See methods section for more details about the MRPP statistics.

| Sample Year 1 | Unburned Plots | | | Burned Plots | | |
|---|---|---|---|---|---|---|
| | **T** | **A** | ***p*** | **T** | **A** | ***p*** |
| Overall | −6.186 | 0.305 | **<0.001** | −8.063 | 0.162 | **<0.001** |
| PI vs. PII | −1.522 | 0.060 | 0.079 | −1.778 | 0.039 | 0.054 |
| PI vs. PIII | −5.532 | 0.383 | **0.001** | −6.360 | 0.124 | **<0.001** |
| PI vs. Mature | −5.958 | 0.360 | **0.001** | −7.696 | 0.240 | **<0.001** |
| PII vs. PIII | −4.439 | 0.246 | **0.003** | −5.048 | 0.107 | **0.001** |
| PII vs. Mature | −4.207 | 0.215 | **0.004** | −6.186 | 0.207 | **0.001** |
| PIII vs. Mature | 0.399 | −0.020 | 0.550 | −1.222 | 0.024 | 0.112 |
| Sample Year 2 | T | A | *p* | T | A | *p* |
| Overall | −7.448 | 0.294 | **<0.001** | −7.570 | 0.144 | **<0.001** |
| PI vs. PII | −3.900 | 0.144 | **0.005** | −0.591 | 0.011 | 0.240 |
| PI vs. PIII | −5.591 | 0.387 | **0.002** | −8.418 | 0.158 | **<0.001** |
| PI vs. Mature | −6.187 | 0.358 | **0.001** | −7.837 | 0.233 | **<0.001** |
| PII vs. PIII | −4.551 | 0.167 | **0.003** | −4.297 | 0.077 | **0.004** |
| PII vs. Mature | −4.872 | 0.162 | **0.002** | −5.267 | 0.137 | **0.001** |
| PIII vs. Mature | −0.468 | 0.012 | 0.306 | 0.311 | −0.005 | 0.515 |

According to expectation, differences in plant community composition were observed between unburned and burned plots within each woodland development phase for both sampling periods. The differences were more pronounced in the advanced woodland development stages (A = 0.409, *p* < 0.001 for sampling period 1 and A = 0.415, *p* < 0.001 for sampling period 2 in Mature plots) compared to Phase I (A = 0.083, *p* = 0.015 for sampling period 1 and A = 0.160, *p* = 0.040 for sampling period 2), indicating longer-lasting effects of wildfire in the more advanced woodland phases.

Differences in plant community composition were observed in all woodland development phases between sampling periods in the burned plots, indicating continued successional development in the plots. However, the A-values were relatively low (0.046–0.098), indicating moderate ecological differences, which could be expected given that the samples were taken only seven years apart in a semi-arid ecosystem. As expected, no difference in plant community composition was observed between time periods for unburned plots in any phase.

### 3.3. Functional Groups

Changes in plant cover by functional group were summarized by time period and woodland development phase, and differences between unburned and burned plots were evaluated statistically.

Annual grass cover was low across unburned control plots and burned plots for all woodland development phases for both sampling periods. No significant difference was detected when unburned and burned plots of the same phase and time period were compared (Figure 6). Frequent annual grasses were cheatgrass and soft brome. Generally, cheatgrass cover was less than 0.1% with occasional plots up to 2% (Table S1). Annual grasses increased slightly after the burn, particularly in developed woodlands; 5–6 years after fire, cheatgrass cover was 1.8 ± 4.0% in woodlands that were in Phase III pre-fire and 0.8 ± 1.4% in woodlands that were in the Mature class pre-fire (Table S1). By the second

sampling period, cheatgrass had decreased to $0.3 \pm 0.5$ in the burned Phase III woodlands and to $0.5 \pm 0.4$ in the burned Mature woodlands. Soft brome was present on many plots at low cover levels (0–1%).

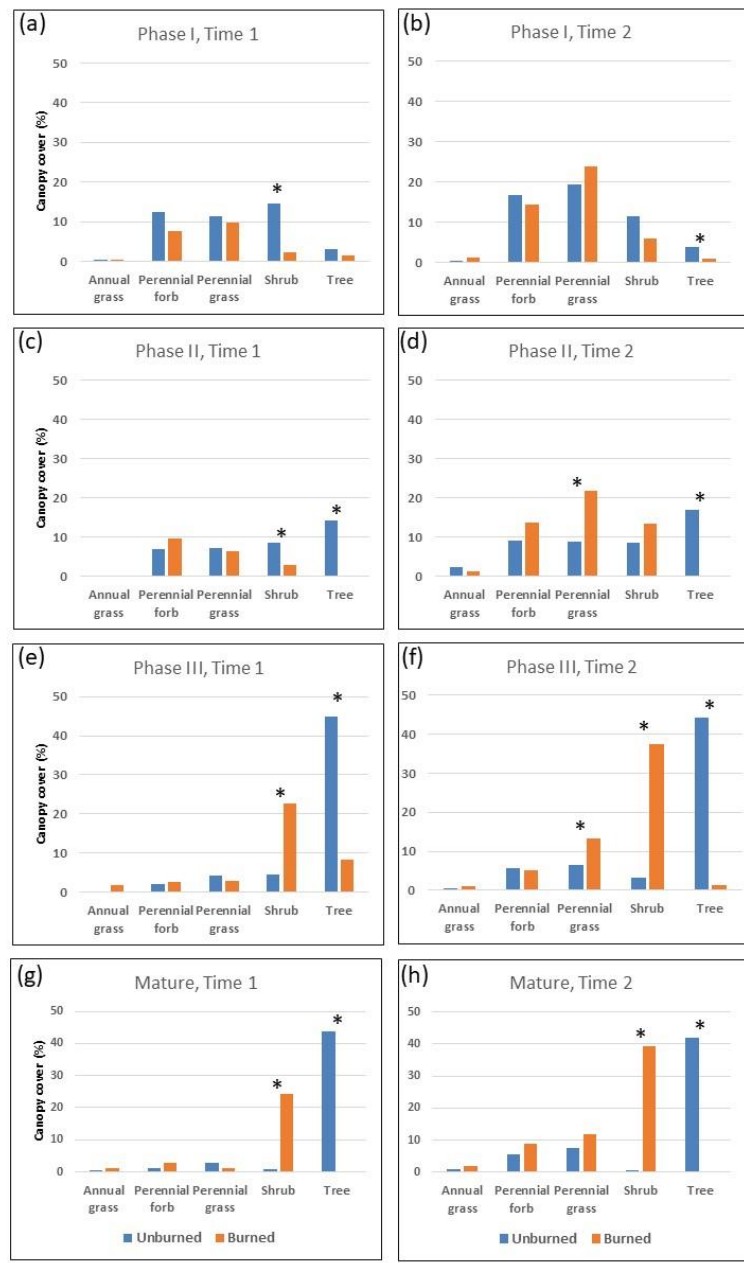

**Figure 6.** Comparison of canopy cover of functional plant groups between burned and unburned plots by woodland development phase and sampling period. Significant differences ($p < 0.05$) in cover between burned and unburned plots are marked with a star (*).

Perennial forb cover was not significantly affected by the wildfire when compared within woodland development phase for either sampling period. Perennial forbs were around 10% cover in Phase I in the first sampling period and around 15% in the second sampling period. Decreasing cover levels of perennial forbs were observed along the successional gradient. In Phase II woodlands, the perennial forb cover was around 8% during the first sampling period and around 10% in the second sampling period. In Phase III woodlands, perennial forb cover was <5% and varied in Mature woodlands (2–9% cover).

Perennial grass cover was not affected by the wildfire in any woodland phase in the first sampling period. However, during the second sampling period, perennial grass cover was significantly higher on burned plots of Phase II (Figure 6d) and Phase III (Figure 6f). Overall, perennial grass cover was about 10% in both unburned and burned Phase I woodlands in sampling period 1 and around 20% in sampling period 2. In burned Phase II woodlands, the perennial grass cover was around 7% in sampling period 1 and over 20% by sampling period 2, a significant increase compared to unburned plots. In Phase III woodlands, perennial grasses were lower, around 5%, but increased to over 10% by the second sampling period. Trends in Mature woodlands were similar to those in Phase III, but no significant differences were documented.

Shrub cover was significantly lower in burned Phase I woodlands compared to unburned, indicating that shrubs had not returned to pre-fire cover levels after 5–6 years (Figure 6a) while 12–13 years post-fire, shrub cover was approaching pre-fire cover levels, and the difference in shrub cover was no longer significant between burned and unburned plots (Figure 6b). Shrub cover in unburned Phase I plots was dominated by mountain big sagebrush (13.4 $\pm$ 9.0%; Table S1), while burned plots had a higher diversity of shrub species. Mountain big sagebrush cover was still low in burned Phase I plots (3.9 $\pm$ 9.0%; Table S1), but sprouting shrubs (e.g., rabbitbrush and bitterbrush) and seedbank shrubs (e.g., snowbrush ceanothus) had increased. In Phase II woodlands, significant reductions of shrub cover was documented because of the fire in the first sampling period (Figure 6c) with no discernable differences in the second sampling period. Similar to unburned Phase I woodlands, mountain big sagebrush was the dominant shrub in unburned Phase II woodlands (5.9 $\pm$ 4.3%; Table S1), while the burned Phase II woodlands had low shrub cover in sampling period 1, but by sampling period 2, shrub cover had increased to almost 14% on average and was composed of a variety of shrubs including mountain big sagebrush (2.4 $\pm$ 3.1%; Table S1), snowbrush ceanothus (8.3 $\pm$ 10.7%; Table S1) and small amounts of other shrubs including rabbitbrush, bitterbrush, mountain snowberry and wax currant (*Ribes cereum* Douglas). Shrub cover in unburned Phase III and Mature woodlands was low (<4%) and composed of small amounts of mountain big sagebrush (<1%), curl-leaf mountain mahogany, rabbitbrush, bitterbrush and mountain snowberry; notably, snowbrush ceanothus was not detected along the transects in unburned plots but occasionally observed in the general area.

The only tree species present on the plots was western juniper. Juniper cover decreased within all woodland phases as a result of the wildfire as expected, with a more pronounced change in the later woodland development phases (Figure 6). Juniper cover pre-fire was <5% in Phase I woodlands, around 15% in Phase II woodlands, and in Phase III and Mature woodlands, juniper cover was over 40%. Juniper trees were still absent after 12–13 years in burned areas (Figure 6b,d,f,h). Juniper seedlings will be discussed later.

Annual forbs were not analyzed for change in cover because of small cover values (<1%) across plots, which are difficult to accurately assess in the field. Common annual forb results are therefore reported in terms of frequency later in the manuscript.

*3.4. Changes in Cover of Dominant Species*

Cover of the 14 most dominant species in the dataset are reported by woodland development phase, burn status and sampling period in Table S1. Dominant perennial forbs were arrowleaf balsamroot, tapertip hawksbeard and lupine. They were present in all woodland development phases during both sampling periods and in unburned and burned plots. Generally, higher cover levels (up to 6.3% on average) were present in earlier woodland phases. By the second sampling period arrowleaf balsamroot was higher in burned than in unburned Phase I woodlands. Tapertip hawksbeard was higher in all woodland development phases in the second sampling period in burned plots and was significantly higher in burned compared to unburned plots, with cover levels around 2% on average. Lupine did not show a significant change in either sampling period or woodland

phase but tended to be higher (2–5%) in early woodland phases and lower in late woodland phases (<1%).

Dominant perennial grasses included Columbia (*A. nelsonii* [Scribn.] Barkworth) and western needlegrass, bottlebrush squirreltail (*Elymus elymoides* [Raf.] Swezey), Idaho fescue, Sandberg bluegrass and bluebunch wheatgrass. These grasses were present in all unburned or burned woodland development phases in both sampling periods. Overall, these perennial grasses tended to increase over time and respond positively to the burn, except Idaho fescue, which was lower in burned compared to unburned Mature woodlands. Columbia needlegrass cover was on average 0.2–1.9% across unburned and burned plots. Columbia needlegrass was significantly higher in burned Phase II, Phase III and Mature plots in the second sampling period and was also higher in burned compared to unburned Phase II plots. No significant changes were observed for western needlegrass, which was present at 0.1–1.7% on average across unburned and burned woodland phases. Bottlebrush squirreltail was significantly higher in burned Phase II, Phase III and Mature plot in the second sampling period and was also higher in burned Phase III and Mature plots compared to unburned plots. Idaho fescue was higher in the second sampling period in unburned Phase I and Mature plots and in burned Phase I and II plots. Sandberg bluegrass was higher in the second sampling period in unburned Phase I plots and in burned Phase I, II and III plots. Sandberg bluegrass was higher in burned Phase III and Mature plots compared to unburned plots of the same phases. Bluebunch wheatgrass was higher in burned Phase I and II plots compared to unburned plots and increased over time in these phases.

Dominant shrub species included mountain big sagebrush, snowbrush ceanothus and mountain snowberry. Mountain big sagebrush was unsurprisingly lower in burned Phase I woodlands compared to unburned, and mountain snowberry was lower in burned Phase III woodlands compared to unburned. Snowbrush ceanothus was higher in burned Phase III and Mature woodlands compared to unburned and increased significantly over time in those woodland phases. Snowbrush ceanothus was not present in unburned plots of any phase but by the second sampling period was very dominant in burned Phase III (35.0 ± 27.3%) and Mature woodlands 38.0 ± 20.3%).

The only tree species present in the area was western juniper. The juniper cover in Phase I was low prior to the fire (3–4% on average) and increased along the successional gradient to 10–20% in Phase II and over 40% in Phase III and in the Mature woodlands. Juniper decreased to close to 0% in Phase II, Phase III and Mature woodlands because of the high fire severity. A couple of Phase III plots at the edge of the fire burned at low severity, and in these plots a few large juniper trees survived.

Cheatgrass was absent in unburned plots in the first sampling period but present at low levels (~1%) during the second sampling period. No significant changes in cheatgrass were detected along the successional gradient or because of the wildfire (Table S1).

### 3.5. Species with High Frequency but Low Cover

Species with low cover were compared using frequency rather than cover because it is very difficult to assign accurate cover values to species with low cover (<1%). Frequent species are listed in Table 3. Frequent annual forbs with low cover included *Collomia linearis*, *Collinsia parviflora*, *Cryptantha* spp., *Epilobium brachycarpum*, *Lactuca serriola* and *Stellaria* spp. These annual forbs occurred frequently across woodland development phases and in both burned and unburned areas. A few trends can be observed. *Collomia linearis* occurred across unburned and burned plots of all woodland phases, with higher frequency the second sampling period. *Collinsia parviflora* frequency tended to be lower in burned plots and also decreased between time periods. *Cryptantha* spp. frequency was higher in burned plots but decreased over time; however, even in the second sampling period, the frequency was higher in burned plots. *Epilobium brachycarpum* and *Lactuca serriola* frequency was higher in burned plots compared to unburned, and *L. seriola* tended to decrease over time. *Stellaria* was higher in burned plots the first sampling period but had decreased by the second sampling period.

**Table 3.** Mean frequency (%) and standard deviation in parenthesis for species that occurred frequently in sampled quadrats but at low percent cover.

| Period | Phase | Burn | N | dNBR 1-yr | *Bromus tectorum* | *Collomia linearis* | *Collinsia parviflora* | *Crepis acuminata* | *Cryptantha spp.* | *Epilobium brachty-carpum* | *Lactuca serriola* | *Phacelia heterophylla* | *Phlox longifolia* | *Stellaria spp.* | *Tragopogon dubius* | *Viola purpurea* |
|---|---|---|---|---|---|---|---|---|---|---|---|---|---|---|---|---|
| 1 | 1 | 0 | 5 | −36 (11) | 0.0 | 76.0 (15.5) | 49.2 (9.7) | 22.8 (17.5) | 8.4 (18.8) | 5.6 (11.4) | 0.0 | 0.0 | 46.4 25.5) | 47.6 (24.6) | 1.6 (2.6) | 10.8 (11.4) |
| 1 | 1 | 1 | 7 | 47 (93) | 11.4 (18.8) | 60.9 (10.0) | 35.1 (22.4) | 51.1 (20.3) | 24.3 (16.1) | 33.1 (36.0) | 12.6 (14.6) | 0.0 | 53.7 (14.9) | 65.1 (21.9) | 4.6 (4.6) | 7.1 (8.8) |
| 1 | 2 | 0 | 5 | −23 (35) | 0.0 | 53.6 (49.9) | 48 (48.4) | 36.8 (33.4) | 3.6 (3.1) | 8.0 (9.6) | 0.0 | 0.0 | 44.8 (42.6) | 60.0 (62.0) | 0.0 | 15.2 (12.6) |
| 1 | 2 | 1 | 5 | 139 (123) | 4.7 (8.2) | 51.0 (25.3) | 37.7 (19.4) | 46.0 (10.9) | 53.3 (16.1) | 29.3 (22.1) | 19.3 (18.3) | 8.7 (17.4) | 32.7 (20.7) | 73.7 (14.6) | 8.3 (6.3) | 3.0 (2.1) |
| 1 | 3 | 0 | 5 | −20 (12) | 0.0 | 27.2 (22.3) | 52.4 (32.1) | 5.2 (6.4) | 5.6 (6.7) | 0.4 (0.9) | 0.0 | 0.8 (1.8) | 10.4 (10.9) | 49.6 (13.0) | 0.0 | 32.8 (10.6) |
| 1 | 3 | 1 | 15 | 284 (16) | 21.7 (25.3) | 19.9 (16.0) | 3.0 (23.3) | 10.0 (13.7) | 50.0 (28.5) | 27.6 (25.7) | 36.9 (29.8) | 18.7 (18.3) | 6.3 (8.0) | 61.5 (15.9) | 7.2 (9.9) | 8.5 (7.5) |
| 1 | 4 | 0 | 5 | −12 (17) | 11.7 (15.1) | 38.7 (14.8) | 76.0 (11.8) | 8.7 (12.2) | 5.3 (4.3) | 4.3 (5.0) | 0.3 (0.8) | 1.3 (1.6) | 10.7 (9.4) | 50.3 (32.3) | 0.0 | 40.0 (15.4) |
| 1 | 4 | 1 | 8 | 396 (74) | 23.5 (20.5) | 19.8 (16.5) | 23.5 (19.9) | 25.5 (17.3) | 77.3 (14.8) | 67.5 (20.2) | 47.8 (29.7) | 30.8 (22.9) | 14.5 (11.1) | 49.8 (26.8) | 5.0 (4.4) | 5.8 (5.7) |
| 2 | 1 | 0 | 7 | −36 (11) | 2.8 (5.2) | 57.2 (19.6) | 9.2 (3.0) | 24.8 (15.4) | 12.8 (13.2) | 12.0 (10.8) | 0.4 (0.9) | 0.0 | 57.2 (23.8) | 23.2 (14.5) | 0.4 (0.9) | 12.8 (10.0) |
| 2 | 1 | 1 | 7 | 47 (93) | 10.3 (14.3) | 63.7 (15.3) | 14.0 (13.0) | 47.1 (14.3) | 23.1 (17.2) | 54.3 (29.4) | 2.6 (4.3) | 0.3 (0.8) | 54.0 (15.6) | 10.3 (10.5) | 17.7 (19.2) | 12.0 (13.1) |
| 2 | 2 | 0 | 5 | −23 (35) | 1.2 (1.1) | 37.2 (27.5) | 23.6 (15.1) | 18.0 (14.8) | 17.6 22.7) | 9.6 (10.4) | 0.0 | 1.2 (1.1) | 33.2 (23.9) | 47.2 (20.5) | 0.4 (0.9) | 28.4 (16.9) |
| 2 | 2 | 1 | 5 | 139 (123) | 11.3 (18.4) | 59.3 (16.5) | 9.0 (10.5) | 58.0 (11.0) | 31.0 (20.1) | 43.3 (14.1) | 7.7 (9.4) | 3.7 (9.0) | 40.7 (12.6) | 11.3 (9.3) | 22.7 (10.4) | 7.3 (6.2) |
| 2 | 3 | 0 | 5 | −20 (12) | 4.8 (7.8) | 34.8 (31.3) | 56.0 (15.0) | 2.0 (2.8) | 11.6 (9.0) | 17.2 (15.0) | 1.2 (1.8) | 6.8 (8.3) | 11.2 (11.2) | 47.6 (28.8) | 0.0 | 53.2 (6.6) |
| 2 | 3 | 1 | 14 | 284 (165) | 22.3 (18.8) | 52.6 (17.3) | 9.3 (10.2) | 24.9 (17.8) | 28.1 (15.0) | 36.0 (26.0) | 10.7 (8.4) | 21.6 (19.8) | 11.9 (13.3) | 17.9 (12.7) | 32.9 (19.5) | 8.6 (6.3) |
| 2 | 4 | 0 | 5 | −12 (17) | 19.2 (20.5) | 35.6 (19.1) | 31.2 (17.9) | 11.2 (9.2) | 12.0 (9.9) | 11.6 (6.2) | 1.2 (1.1) | 4.4 (5.4) | 19.6 (15.6) | 57.2 (22.5) | 0.4 (0.9) | 43.6 (16.8) |
| 2 | 4 | 1 | 8 | 396 (74) | 39.0 (21.3) | 49.5 (21.7) | 6.0 (4.4) | 27.5 (17.1) | 32.8 (12.7) | 37.8 (17.0) | 13.8 (9.0) | 38.5 (27.9) | 13.5 (12.1) | 18.3 (15.6) | 42.0 (27.6) | 7.5 (10.1) |

Frequent perennial forbs included Crepis acuminata, Phacelia heterophylla, Phlox longifolia and Tragopogon dubius. *C. acuminata* and *P. heterophylla* increased with fire, particularly in the second time period. *P. longifolia* decreased with fire. The frequency of Bromus tectorum increased with fire.

### 3.6. Juniper Seedlings

Juniper seedlings were counted within the quadrats along the transects on burned plots and unburned control plots. The number of seedlings was zero or near zero on both burned and unburned plots when the pre-fire juniper cover was less than 10%, but as pre-fire juniper cover increased, more seedlings were observed, particularly on unburned plots (Figure 7a). Significantly higher density of juniper seedlings was found on unburned plots for Phase II, while significant differences between burned and unburned plots were not detected in Phase I, III or Mature woodlands (Figure 7b). Lack of difference in Phase III and Mature woodlands is likely due to high variability in the seedling density in unburned woodlands.

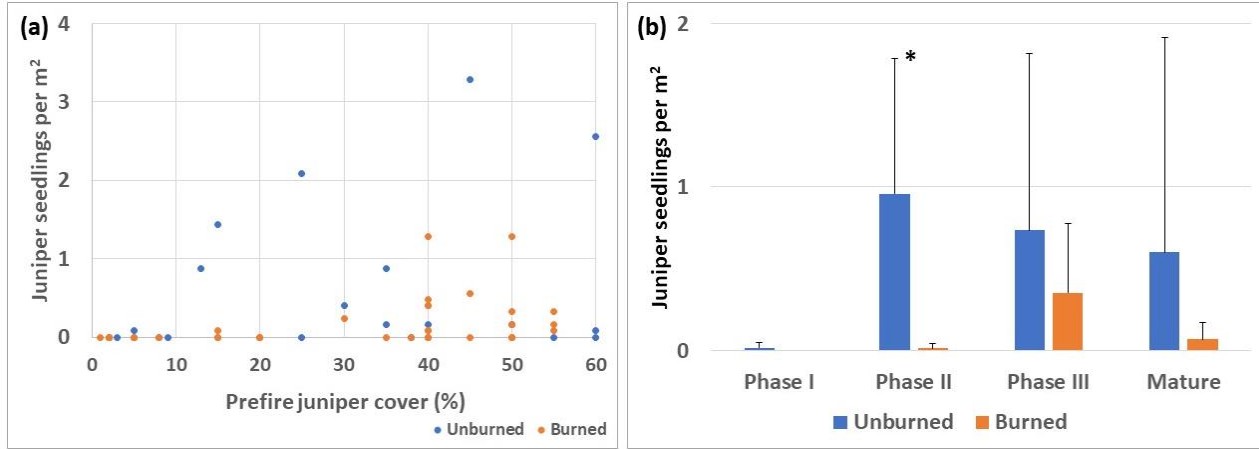

**Figure 7.** Juniper seedlings per m$^2$ versus pre-fire percentage juniper cover for unburned and burned plots (**a**) and by woodland development phase (**b**). Error bars represent standard deviation, and significance ($p < 0.05$) is denoted with a star (*). Both graphs reflect data from sampling period 2 (12–13 years post-fire); no juniper seedlings were observed along the transects on burned plots in sampling period 1.

### 3.7. Successional Pathways

Our research suggests that late successional plots that burn at high severity may not necessarily return to sagebrush steppe vegetation but rather may follow a different sere such as one dominated by snowbrush ceanothus during the pathway back to mature juniper woodland (Figure 8). The snowbrush ceanothus pathway may occur in areas where climate conditions are suitable for the species, and snowbrush seeds are present in the seedbank. In Idaho, snowbrush ceanothus has been reported to occur across a wide elevation range (1150–3000 m) and annual precipitation range (690–1140 mm) [44], and in Utah, snowbrush ceanothus has been observed at lower precipitation ranges (410–510 mm; [45]).

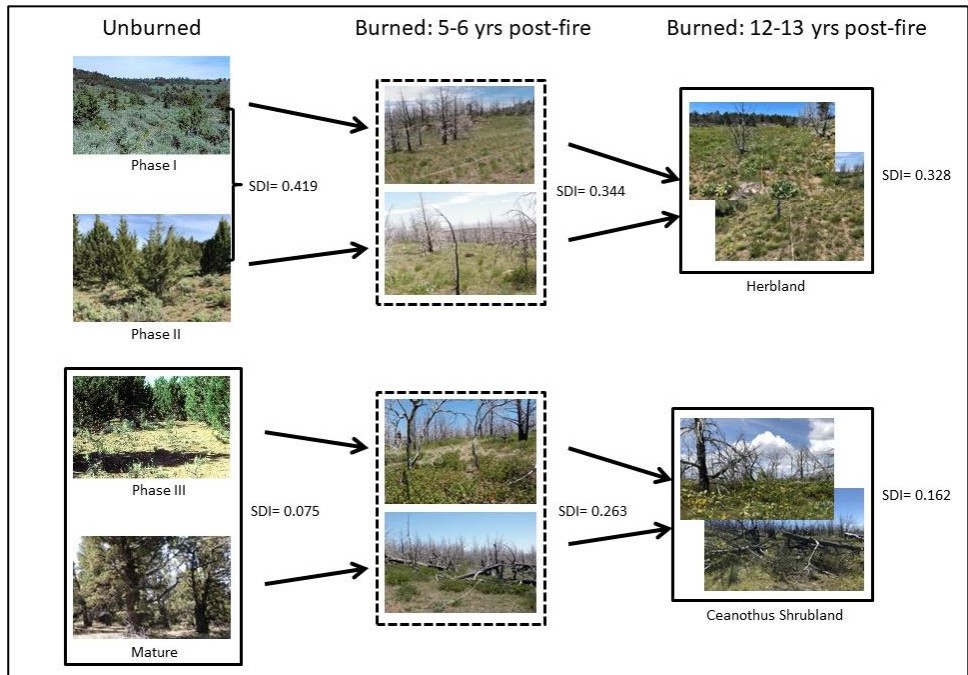

**Figure 8.** Proposed post-fire successional pathways as influenced by pre-fire plant community composition and structure. Early successional stages (Phase I and II) result in a herb-dominated community 12–13 following fire. Sorensen's Dissimilarity Index (SDI) indicates dissimilarity between phases and ranges from 0 to 1, where 0 means the plots are identical, and 1 means no similarity (complete dissimilarity). Pre-fire, plant communities of Phase I and II plots are somewhat similar (SDI = 0.419), while Phase III and Mature plots are very similar (SDI = 0.075). Following fire, burned Phase I and II plots become more similar (SDI = 0.344 after 5–6 years, and SDI = 0.328 after 12–13 years). Burned Phase III and Mature plots also stay similar as succession progresses (SDI = 0.263 after 5–6 years, and SDI = 0.162 after 12–13 years). The dashed-line box indicates that the communities within the box are not significantly different from each other in spite of being initially different pre-fire.

## 4. Discussion

### 4.1. Richness and Diversity

Average richness of vascular plant species decreased slightly along the successional gradient in unburned plots, which was also reflected in burned plots during the first sampling period, but this decrease was not observed in the second sampling period. The average richness per sampled plot remained fairly constant along the sere and in both unburned and burned plots, at 28–35 species 5–6 years post-fire and 35–40 species 12–13 years post-fire. Species richness increased between the two sampling periods across woodland development phases, except in Phase II. We attribute this change to successional development within the plots. Diversity (H') was lower in late successional phases (Phase III and Mature) compared to the early successional phases (Phase I and II) in both unburned and burned plots. Diversity accounts for both changes in richness and relative abundance, and since richness did not change much along the sere, we attribute the decrease in diversity to the increased dominance of juniper trees in unburned late-successional plots and to the dominance of snowbrush ceanothus in burned plots that were late-successional pre-fire. Diversity increased over time in burned Phase III plots but not in earlier successional phases. The increase in diversity can be attributed to the documented increased richness as the plots began to recover from the burn.

We also evaluated the contribution of richness and diversity of each woodland development phase to the richness and diversity of the entire landscape. For this analysis, we computed the sum of all species in plots of each phase rather than averages. Species richness was in the 46–74 range across phases for burned and unburned plots; however,

total species richness for the landscape (all plots) was double that, 124 in sampling period 1 and 131 in sampling period 2 (Figure 4a,b). The higher values for the total landscape can be explained by species turnover resulting from successional development and from the burn. Earlier successional phases, however, contribute more to the vascular plant diversity of the landscape compared to later successional stages (Figure 4c,d). In fact, species diversity of the early successional phases is equal to or higher than the species diversity of the landscape. We can also conclude that fire contributed to increased vascular plant diversity given that diversity is higher in burned plots when compared to unburned plots of the same phase. These data emphasize the importance of maintaining early successional habitats for maximizing vascular plant diversity, but also confirm that all woodland development phases contribute to species richness in the landscape. It has been demonstrated that the changes in plant community composition and vegetation structure cascade into changes in other taxa, birds for example. Pavlacky and Anderson [46] observed richness and diversity in the bird community along successional and elevational gradients in Utah juniper woodlands, concluding that all woodland successional stages contribute to bird gamma diversity in these woodlands. Bird surveys following juniper removal projects have demonstrated that juniper removal increases abundance of nesting pairs of several ground- and shrub-nesting species, while species associated with woodland vegetation decline [47]. These data emphasize the need of having multiple plots of a variety of successional stages present in the total landscape to provide habitat for all species.

### 4.2. Plant Community Turnover and Composition

Even though no differences in species richness or diversity were detected between unburned and burned plots of the same phase in any of the time periods sampled, the burn resulted in changes in species composition demonstrated by the Sorensen Dissimilarity Index (SDI). The difference in species composition (tested with MRPP) between unburned Phase I and II plots was significant in sampling period 2 but not in sampling period 1. The species turnover indicated by an SDI of 0.419 was also relatively low, suggesting strong similarity between Phase I and II woodlands. The largest dissimilarity in species composition (largest turnover of species) in unburned plots was seen between woodland development Phase II and III. The transition from Phase II to Phase III woodlands is the stage in succession when woodland species become dominant over sagebrush steppe species, as previously described by Miller et al. [4]. Strand et al. [22] also documented that the wildfire burn severity was significantly higher in Phase III plots compared to Phase I or II plots in the TCWFC. Weiner et al. [18] showed that the increase in litter and duff ground fuels that have the potential to smolder for long periods of time after the fire front has passed could be one mechanism that explains the higher burn severity in the late successional woodlands. Obviously, the heat pulse from torching juniper crowns also contributes heat to the soil and seed bank that may sustain the burning ground fuels and contribute heat that consumes the organic soil layer, seed banks and other regenerative plant structures in the soil. No difference in species composition between Phase III and Mature woodlands was detected (MRPP analysis) and the dissimilarity between the plots was very low (SDI = 0.075). We interpret these similarities between Phase III and Mature woodlands as a support for the statement that Phase III woodlands have passed a threshold for when ecological processes are dominated by the juniper rather than the steppe vegetation as suggested [4]. Additional support for the suggestion that an ecological threshold has been passed in Phase III woodlands is the dissimilarity observed between unburned and burned plots along the successional gradient. The dissimilarity between unburned and burned plots of Phase I is 0.419, then 0.643, 0.745 and 0.945 for Phase II, III and Mature plots, respectively. The very high dissimilarity between unburned and burned plots (0.945) indicates near complete species turnover since max SDI = 1 for complete turnover in species between plots.

### 4.3. Functional Groups and Common Species

Increases in flammable exotic annual grasses following fire in sagebrush steppe and juniper woodlands has been documented and is a concern. However, we did not detect a significant difference in annual grass cover comparing unburned and burned plots. Cheatgrass cover was absent or low (<1%) in unburned plots, but higher values were detected after fire, with the largest increase in Phase III woodlands (1.8 ± 4.0%; Table S1), although the increase was not significant at $p = 0.05$. Frequency of cheatgrass was also higher in burned plots (10–40%) compared to unburned (0–10%), see Table 3. We do not anticipate that cheatgrass will become dominant in the burned area since the low cover levels have remained low for more than 10 years after the fire. Variability in annual grass response after fire in western juniper woodlands has been reported in the literature. A study by Weiner et al. [18] documented high cover of cheatgrass (9.7 ± 22%) under burned tree canopies six years post-fire in the TCWFC. The highest cover of cheatgrass occurred in areas that burned at low severity with very little cheatgrass in the high-severity areas, possibly because the organic layer of the soil was consumed by the fire, including the cheatgrass seedbank. The sampled plots within the TCWFC were situated on an ecological site classified as high resistance to annual grass invasion because of its higher precipitation zone and soil moisture and temperature regime. On plots with lower resistance and warmer and dryer soil moisture regimes, effects of wildfire on annual grasses could have been very different, with risk of annual grass dominance and initiation of an annual grass–fire cycle.

Several annual forbs occurred frequently across unburned and burned plots throughout both sampling periods including *Collomia linearis*, *Collinsia parviflora*, *Cryptantha* spp., *Epilobium brachycarpum*, *Lactuca seriola*, *Agoseris* Raf. spp. and *Stellaria* spp. These species establish quickly after fire, but little else is known about the ecology of them. We found it interesting that these forbs were frequent across burned areas and throughout the sere in unburned areas. Forbis [48] suggested that native annual forbs are phenologically similar to cheatgrass and therefore may use similar resource pools. Leger et al. [49] documented such competition in greenhouse experiments for selected forbs. For example, *Amsinckia tessellata* A. Gray reduced *B. tectorum* biomass by 97%, and *Amsinckia intermedia* Fisch. and C. A. Mey., *Amsinckia tessellate* and *Descurainia pinnata* (Walter) Britton reduced seed output between 79 and 87%. Further research is needed to better understand the ecology of these frequent and persistent annual forbs.

Perennial grasses and forbs were more abundant in early successional stages when compared to late successional stages in unburned plots. Several other studies have documented this decline in understory herbaceous perennial vegetation along the juniper woodland development gradient [1,4,7,50]. No difference in perennial grass cover was documented between unburned and burned areas 5–6 years post-fire, but by the second sampling period, 12–13 years post-fire, perennial grass cover was higher in burned plots compared to unburned plots for Phase II and III woodlands. Bottlebrush squirreltail, Columbia needlegrass and bluebunch wheatgrass all increased post-fire. However, Idaho fescue was lower in burned Mature plots compared to unburned even after 12–13 years, which can be expected given the high severity burn in Mature plots. Idaho fescue is relatively sensitive to fire because the budding areas are located above or at the soil surface and are therefore more sensitive to fire compared to bluebunch wheatgrass for example [51]. No significant differences were detected in perennial forb cover when unburned plots were compared to burned plots of the same woodland phase. Bates et al. [7] similarly found no difference in perennial forb yield when comparing prescribed burn plots to unburned control plots; however, they observed an increase in tall perennial forbs and a decrease in mat-forming forbs after the burn. It is important to recognize that response to fire is different for each individual forb species as described by Pechanec et al. [52] who classified forbs common to sagebrush steppe by susceptibility to fire damage.

The shrub community was significantly different between unburned and burned plots in all woodland development phases after 5–6 years. In the early unburned woodland phases, big sagebrush was the dominant shrub. However, fire is lethal to mountain big

sagebrush, and the shrub tends to be slow to return to the site after fire because it does not sprout from root or crown, the seed is short-lived in the seed bank [53] and seeds do not spread more than 3 m from shrubs in nearby unburned areas [54,55]. In the years following the fire, mountain big sagebrush cover increased slowly and was $3.9 \pm 9.0\%$ in burned Phase I woodlands and $2.4 \pm 3.1\%$ in Phase II woodlands 12–13 years post-fire. Innes and Zouhar [56] synthesized literature reporting that it may take 20–26 years for sagebrush cover to return to pre-burn levels on mountain big sagebrush sites. Shrub cover of sprouting shrubs such as rabbitbrush and bitterbrush increased more rapidly, overall increasing the shrub diversity in burned early successional plots. Increased shrub diversity with a similar species composition has previously been reported following prescribed fire in mountain big sagebrush steppe [27]. While shrub cover in early woodland development phases decreased by fire, shrub cover in late woodland successional stages increased. Average shrub cover in late woodland development phases was low in unburned plots (0–4%) composed of various mixes of mountain big sagebrush, rabbitbrush, antelope bitterbrush, curl-leaf mountain mahogany, mountain snowberry and wax currant. In burned late woodland development phases, snowbrush ceanothus was by far the most dominant shrub, with average cover levels approaching 40% canopy cover 12–13 years post-fire. The dramatic increase in snowbrush ceanothus in burned late woodland development phases was a surprise because the shrub was rarely present, and if so, only in small amounts, in unburned plots. Snowbrush ceanothus is native to western North America and has commonly been observed to increase following wildfire or prescribed fire even in areas where it was uncommon or absent prior to fire [57,58]. The seed of snowbrush ceanothus can remain viable for more than two centuries in the soil [59], and it has long been known that the seed requires a heat treatment to germinate [60]. Apparently, snowbrush ceanothus seeds must have been present deep in the seedbank of the late successional woodlands, which is surprising given that most of the organic matter had been consumed by the high-severity fire. Ceanothus was particularly prevalent in areas under the burned tree crowns where duff and litter from the old juniper trees had accumulated for in some cases several centuries [18]. Even though some sagebrush was present in the Phase III and Mature woodlands pre-fire, it is clear that those sites are unlikely to return to sagebrush steppe vegetation but will more likely remain a snowbrush ceanothus shrubland for a period of time, then eventually returning to western juniper dominance.

Juniper seedling counts were low (0.01 seedlings per square meter) in unburned Phase I woodlands but almost 1 seedling per square meter on average in unburned Phase II woodlands. By the time the woodland had transitioned into Phase II, many juniper trees had reached maturity, and we anticipate that seeds were abundant in the seed bank. Phase III and Mature woodlands had 0.75 and 0.5 seedlings per square meter on average, respectively, with high variability around the mean, indicating that seeds were present, and germination was occurring. Wozniak and Strand [61] documented similar seedling counts in Phase I western juniper woodlands with higher counts in later successional stages. Juniper trees were generally killed by the high-intensity fire, including centuries-old trees in Mature plots. Only a few juniper trees survived in plots located on the edge of the fire that burned in low or moderately low severity. The loss of juniper in Phase I plots did not result in a large change in composition since juniper only accounted for up to 10% of the canopy cover in these plots. The loss of juniper in Phase III and Mature plots where pre-fire juniper cover was upward of 40% resulted in major changes in plant community composition. In addition, these plots burned at high severity, which resulted in loss of shrubs and herbaceous vegetation as well. No juniper seedlings were detected on burned plots in the first sampling period, but by the second sampling period, 12–13 years post-fire, juniper seedlings were present on burned plots, but the abundance varied by woodland development phase. No juniper seedlings were counted in burned Phase I woodlands, and only 0.01 seedlings per square meter were counted in burned Phase II woodlands. Seedling counts were higher in burned Phase III and Mature woodlands but still less than half the count observed in unburned woodlands of the same phase. Wozniak and Strand [61] also

documented very low seedling counts 10 years after prescribed burns across woodland development phases, ranging from 0.0001 seedlings per square meter in Phase I woodlands to 0.002 seedlings per square meter in Phase III woodlands. The lower seedling counts reported by Wozniak and Strand [61] can be attributed to the fact that their study design excluded seedlings shorter than 5 cm. Given the fact that juniper seedlings were detected on the site 12–13 years post-fire, this suggests that juniper is likely to return to the area, but the juniper establishment will be slow.

## 5. Conclusions

Little information is available about effects of pre-fire vegetation on post-fire plant communities in western juniper woodlands that burned in high intensity wildfire. Given that the area burned in wildfire is increasing across ecosystems in the western US, this research provides new and timely information. Our research demonstrates that early successional stages of woodland development (Phase I and II) can be resilient to wildfire within ecological sites classified as cool and moist soil temperature and moisture regimes [41,62], given that perennial grasses and forb cover were above that of unburned plots, and sagebrush was increasing in cover. Although annual grasses were present, they were not dominant, but we noted that they increased in frequency. These results are encouraging and suggest that prescribed fire or wildland fire use likely will have positive effects on plant community richness, diversity and resilience to future fires on cool and moist mountain big sagebrush ecological sites.

Plots with greater amounts of pre-fire western juniper cover had lower coverage of most shrub and perennial grass species, with lower species diversity. Greater juniper coverage also resulted in greater fire intensity and severity [22]. Thus, late-successional plots had a diminished pre-fire understory but contained species and habitats such as cavities that were uncommon in other stages. The understory was further reduced by the greater fire intensity and severity resulting in lower species diversity. Much of the available area, 35 and 38%, was occupied by *Ceanothus* 12 years post-fire in the Phase III and Mature plots, respectively.

Perennial grasses were an important functional group in the Phase I and II plots and increased to mean coverages of greater than 20% by the end of the study. Perennial grasses also increased on burned Phase III plots but to a lesser degree than on the earlier successional stages.

The species richness data show the importance of maintaining multiple examples of all successional stages on the landscape in order to provide habitat for all potential vascular plant species. The species richness at the landscape scale was approximately twice that of any single successional stage. Similar results have been shown for birds and small mammals [46]. A key issue in this consideration is the management and protection of mature woodlands because they require very long time periods to develop (>500 years). Although they do not commonly burn, this study demonstrates that they can burn under extreme fire conditions. Consequently, some Phase III stands should be allowed to develop into mature woodlands. This requires very long-term management strategies.

**Supplementary Materials:** The following supporting information can be downloaded at: https://www.mdpi.com/article/10.3390/fire6040141/s1, Table S1: Table of mean % cover and standard deviation (in parenthesis) for the 14 most common species in the dataset..

**Author Contributions:** Authors E.K.S. and S.C.B. have equally contributed to the manuscript including original research idea, experimental design, data collection, analysis, writing and editing. All authors have read and agreed to the published version of the manuscript.

**Funding:** This research was partially funded by the U.S. Forest Service Rocky Mountain Research Station grant number 12-JV-11221637-136.

**Institutional Review Board Statement:** Not applicable because this study did not involve humans or animals.

**Data Availability Statement:** Data are available by contacting the corresponding author.

**Acknowledgments:** We would like to thank Penny Morgan, Emerita, Fire Ecology, University of Idaho, and undergraduate and graduate students who helped with field sampling. We thank Steve Jirik, Post Fire Recovery & Noxious Weed Program Lead, Bureau of Land Management, Idaho State Office for review of an early draft of the manuscript.

**Conflicts of Interest:** The authors declare no conflict of interest.

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
