# Peer review of "Effects of Pre-Fire Vegetation on the Post-Fire Plant Community Response to Wildfire along a Successional Gradient in Western Juniper Woodlands"

_fire, doi:10.3390/fire6040141_

Round 1
Reviewer 1 Report
Effects of pre-fire vegetation on the post-fire plant community response to wildfire along a successional gradient in western juniper woodlands
Eva K. Strand and Stephen C. Bunting.
This is an interesting paper because it is interpreting wildfire as an ecological process which affects seral stages of Juniper woodland development somewhat differently, and points to some potential strategies to regenerating older stands following wildfires. Although widespread throughout the Western US, less attention on these savannah-like systems than forests in the west.
Major comments:
It seems one of the important findings here is the abundance of Juniper seedlings that occur in the different seral stages (I, II, III, and woodland) because this is the long-lived species that has important relevance for long-term fire regimes. Most of the annual and perennial herbaceous species and grasses are fairly widespread throughout the west, so are of perhaps less ecological importance. This is presented at the end of the Results section (Figure 7), but authors could make this much clearer by reversing the order of presentation in the Results and Discussion sections.
It would also be helpful to elevate the conceptual model in Figure 7 (it should be Figure 8) to much earlier in the Discussion section or at the end of the Results. This helps tie the analyses in the Results together, rather than at almost the end of the Conclusions section.
The authors indicate that long-term management strategies for preservation of the older seral stages of Western Juniper exist. How does this conflict/contrast with Juniper removal to restore sagebrush dominated areas? It would be interesting to add a bit more about how these two goals could be implemented, given increased wildfire occurrence in the Western US, changing climate and precipitation patterns, and grazing use (domestic and native species). The authors should also consider a comparison to regeneration patterns of long-lived woody species in other systems in the Discussion section. Are there commonalities and/or differences with more southerly systems in semi-arid areas? This would be interesting in developing potential strategies for future management.
It would be helpful for readers if the authors could show or estimate the amount of Juniper tree and sapling mortality that occurred in the sampled plots, if the authors have any census data like this (it seems they have worked and published information from this fire in the past). This could also be extended to shrubs, and would be interesting, even if anecdotal. In the absence of biometric data (although remotely sensed NDVI and any LiDAR data would certainly help), how do the authors know which phase of community development they have sampled post fire? They do stratify data by seral stage, but don’t specifically define how they did this. Even basal area of dead trees and saplings would be useful in this regard.
The Methods and Materials section seems a bit out of order, and would benefit from better organization. For example, information on climate, parent material and soils should appear before descriptions of the vegetation and then details of the wildfire.
The authors define “resilience” and “resistance” in the Methods and Materials section, but this is out of place. It should either appear in the Introduction, and combined with information on recovery following fires, or in the Discussion section when discussing the overall ecological importance of this work. The authors never come back to these concepts, but could either at the end of the Discussion section or the Conclusions. It seems very relevant to their results and the conceptual model in the Conclusions section, and also seems relevant to future management strategies.
The Conclusions section is very long, and it would be more effective to distill the key points down and present those, especially as they relate to the regeneration of older seral stages of Western Juniper dominated ecosystems.
Specific comments
In all of the estimated cover data, did the authors detect any sampling bias? Was it always the same person estimating cover, or how did different individuals “calibrate” estimates. This information would lend greater credibility to the extensive use of cover estimates in the manuscript.
Did the authors test data for normality and kertosis? Also, resampling the same plots is essentially a non-independent time series, so that the standard ANOVA analyses should have modified error structure to account for this.
Table 1 format is incorrect, and should be consistent with other Tables.
Figure 2. Can the authors provide a citation for the burn severity index classes (in the legend or text)? It would also be helpful to readers if some of the vegetation features could be identified on the map, and then points to identify the actual location of the plots samples (where they dispersed across the burn or in one localized area, etc.?).
The format of Table 3 is odd, and this table could appear as an Appendix or in Supplemental Material so as not to detract from the main points in the text. In Table 3, it might be better to present the acutal year sampled for “Year”, and “Burn” would be better as a “yes” or “no”. The use of “A” and “B” is confusing, because some letters have “+” and “-“ signs, even for the same letter (B).
Figure 4 legend is incomplete. The authors should explain each graph by letter (a,b,c,d). “c” and “d” are undefined here, but should be.
Authors should check for typographical errors and ensure that all units are metric.
It seems that the Reference list is not in the correct format.
Some of the lengthy text presenting the results of the herbaceous species analyses could be shortened. The authors should also consider presenting some of this information in Supplemental Material. Overall, an interesting message and conceptual model gets a bit buried in the details, and readers would benefit from a clearer presentation and a discussion of potential management strategies, especially given the contrasting and sometime conflicting management activities that currently occur (sagebrush ecosystem restoration, grazing allotments, conservation of “old-growth” Juniper stands, wildfire suppression, fuels management, etc.).
Author Response
Dear Editorial Office,
Please find the response to the 3 reviews in the attached file. We thank the reviewers and the editorial office for constructive feedback and look forward to hearing from you.
Eva Strand

Reviewer 2 Report
- Don’t know if the “PJ” acronym is needed, might read read a little easier if Pinyon-Juniper is spelled out.
- I would suggest specifying the US state(s) and general geographic location of the Tongue-Crutcher Wildland fire complex prior to the methods (i.e., in either abstract or intro).
- Missing commas in this sentence: “reported wind gusts of up to 22 ms−1 temperatures over 38oC and daytime relative humidity below 10%.”
- Maybe try to reformat Table 1, hard to read as is.
- No other comments, very well written paper clearly communicating results.
Author Response
Please see attachement
Reviewer 3 Report
I reviewed the manuscript: Effects of pre-fire vegetation on the post-fire plant community response to wildfire along a successional gradient in western juniper woodlands
By authors: Eva K. Strand and Stephen C. Bunting
The subject manuscript reports long-term results of vegetation dynamics in a western juniper woodland 6 years and 13 years after a wildfire in southwestern Idaho. Overall, I found the manuscript well organized, clearly written, and informative.
My specific comments include:
Line numbering of the manuscript would have been helpful for review comments.
Introduction: Well referenced and presented. Good description of general fire histories and effects in juniper woodlands.
Methods section: In the site description section – what was the fire history of the study site? When was it last burned? As shown in Figure 1, and stated in the text “… total tree cover … usually exceeds 30%.” If data are available, please provide a table of juniper tree numbers and diameter distribution on the unburned control sample area. It is stated that the burned site is phase I development; what was development phase of the control area?
Results section: Adequate, however, in Table 1, the SS column could be dropped because that value can be calculated from df and MS.
Discussion section: I found this section very well presented with adequate references comparing results of this study with those reported elsewhere.
Correction: At the top of page 16, in the paragraph beginning “Cover of the most 14 most dominant … it appears an extra word (most) is present.
Author Response
Please see attachement
